# A Consistent Protocol Reveals a Large Heterogeneity in the Biological Effectiveness of Proton and Carbon-Ion Beams for Various Sarcoma and Normal-Tissue-Derived Cell Lines

**DOI:** 10.3390/cancers14082009

**Published:** 2022-04-15

**Authors:** Masashi Yagi, Yutaka Takahashi, Kazumasa Minami, Taeko Matsuura, Jin-Min Nam, Yasuhito Onodera, Takashi Akagi, Takuya Maeda, Tomoaki Okimoto, Hiroki Shirato, Kazuhiko Ogawa

**Affiliations:** 1Department of Carbon Ion Radiotherapy, Osaka University Graduate School of Medicine, 2-2 Yamada-oka, Suita City 566-0871, Osaka, Japan; k_minami@radonc.med.osaka-u.ac.jp; 2Department of Medical Physics and Engineering, Osaka University Graduate School of Medicine, 1-7 Yamada-oka, Suita City 566-0871, Osaka, Japan; ytakahashi@radonc.med.osaka-u.ac.jp; 3Faculty of Engineering, Hokkaido University, North-13 West-8, Kita-ku, Sapporo 060-8628, Hokkaido, Japan; matsuura@med.hokudai.ac.jp; 4Global Center for Biomedical Science and Engineering, Faculty of Medicine, Hokkaido University, Sapporo 060-8638, Hokkaido, Japan; nam.jinmin.4i@kyoto-u.ac.jp (J.-M.N.); yonodera@med.hokudai.ac.jp (Y.O.); shirato@med.hokudai.ac.jp (H.S.); 5Department of Radiation Physics, Hyogo Ion Beam Medical Center, 1-2-1 Kouto Shingu-cho, Tatsuno City 679-5165, Hyogo, Japan; t.akagi@hibmc.shingu.hyogo.jp; 6Department of Radiology, Hyogo Ion Beam Medical Center, 1-2-1 Kouto Shingu-cho, Tatsuno City 679-5165, Hyogo, Japan; rvks85843@iris.eonet.ne.jp (T.M.); kiamoto935@gmail.com (T.O.); 7Department of Radiation Oncology, Osaka University Graduate School of Medicine, 2-2 Yamada-oka, Suita City 566-0871, Osaka, Japan; kogawa@radonc.med.osaka-u.ac.jp

**Keywords:** particle therapy, carbon ions, proton, radiotherapy, radiobiology, RBE

## Abstract

**Simple Summary:**

Using a consistent experimental protocol, we found a large heterogeneity in the relative biological effectiveness (RBE) values of both proton and carbon-ion beams in various sarcomas and normal-tissue-derived cell lines. Our data suggest that proton beam therapy may be more beneficial for some types of tumors. In carbon-ion therapy, for some types of tumors, large heterogeneity in RBE should prompt consideration of dose reduction or an increased dose per fraction. In particular, a higher RBE value in normal tissues requires caution. Specific dose evaluations for tumor and normal tissues are needed for both proton and carbon-ion therapies.

**Abstract:**

This study investigated variations in the relative biological effectiveness (RBE) values among various sarcoma and normal-tissue-derived cell lines (normal cell line) in proton beam and carbon-ion irradiations. We used a consistent protocol that specified the timing of irradiation after plating cells and detailed the colony formation assay. We examined the cell type dependence of RBE for proton beam and carbon-ion irradiations using four human sarcoma cell lines (MG63 osteosarcoma, HT1080 fibrosarcoma, SW872 liposarcoma, and SW1353 chondrosarcoma) and three normal cell lines (HDF human dermal fibroblast, hTERT-HME1 mammary gland, and NuLi-1 bronchus epithelium). The cells were irradiated with gamma rays, proton beams at the center of the spread-out Bragg peak, or carbon-ion beams at 54.4 keV/μm linear energy transfer. In all sarcoma and normal cell lines, the average RBE values in proton beam and carbon-ion irradiations were 1.08 ± 0.11 and 2.08 ± 0.36, which were consistent with the values of 1.1 and 2.13 used in current treatment planning systems, respectively. Up to 34% difference in the RBE of the proton beam was observed between MG63 and HT1080. Similarly, a 32% difference in the RBE of the carbon-ion beam was observed between SW872 and the other sarcoma cell lines. In proton beam irradiation, normal cell lines had less variation in RBE values (within 10%), whereas in carbon-ion irradiation, RBE values differed by up to 48% between hTERT-HME1 and NuLi-1. Our results suggest that specific dose evaluations for tumor and normal tissues are necessary for treatment planning in both proton and carbon-ion therapies.

## 1. Introduction

Radiation oncology departments worldwide increasingly utilize particle beam therapy, including proton and carbon-ion beams [1,2]. Particle beam therapy has a fascinating physical characteristic—the Bragg peak associated with the maximum energy loss of the particles is located at the region of interest; there is a dramatic reduction in energy release beyond this region [3]. Moreover, there is a rapid fall-off in the dose to nontarget regions, such as normal tissue, which ensures that the beam is accurately focused on the target region. Carbon-ion therapy has several biological benefits, including stronger cell-killing effects than photon beams (even in hypoxic and photon-resistant tumors), strong suppression of endothelial cell migration and the metastatic potential of cancer cells, and enhancement of the antitumor immune response [4,5,6,7,8,9,10,11,12,13,14,15,16].

In particle therapy, dose calculation is performed to estimate the dose distribution in a patient’s body. For prescribing a dose, a spread-out Bragg peak (SOBP) is created [3]. Particle beams are biologically more effective than photon beams, even at the same physical dose, because of the way they transfer energy. Particle beam energy transfer is described as a linear energy transfer (LET). The relative biological effectiveness (RBE) of particle beam irradiation, which is used in current clinical treatment planning systems, can be attributed to LET [17]. For example, in proton therapy, the prescribed dose is commonly calculated from treatment planning systems as the physical dose multiplied by 1.1. RBE values in both tumor and normal tissues have been assumed to be constant. However, there is no evidence to support this assumption, which is an issue that has been ignored.

In carbon-ion therapy, the therapeutic dose is calculated by multiplying the physical dose by the RBE value based on biological and clinical effectiveness [18]. However, this computation is more complicated than that of proton therapy [17]. In carbon-ion therapy, the RBE value depends on LET; therefore, an accurate estimation of the RBE value is essential for obtaining accurate clinical dose distributions. Various analytical and empirical models have been utilized in treatment planning systems to estimate the RBE value of carbon-ion irradiation [19,20]. However, there are uncertainties in the calculation of RBE values due to (i) errors in the estimation of the spatial distribution of RBE values and (ii) errors in experimentally derived absolute RBE values [17]. These uncertainties that are derived from biological factors are characterized as an α- or β-value in linear–quadratic models (LQMs) [21] of cell survival.

Clinical applications that adopt the RBE model, such as the mixed beam [19] and microdosimetric kinetic models [20], are based on experimental results from colony formation assays of the human submandibular gland cell line (HSG). In one of the earliest reports, Kanai et al. demonstrated that the HSG cells have a rather large α- or β-value with a small shoulder; thus, HSG cells mimic the initial reactions and responses of those normal and tumor cells, respectively, that have similar α- and β-values [22]. A number of reports have demonstrated large variations in RBE values in different tumor cell lines. In particular, colony formation assays for various human cell lines showed that the RBE value ranged from 1.06 to 1.32 at an LET of 13.3 keV/µm and from 2.00 to 3.01 at an LET of 77 keV/µm using carbon-ion monobeams [23]. Similar variations in RBE values were also reported in the SOBP [24]. Friedrich et al. established a database of different RBE values for various tumors in various particle beams [25]. However, the data were derived using different experimental protocols at different institutes (e.g., Lawrence Berkeley Laboratory, Gesellschaft für Schwerionenforschung mbH (GSI), and National Institute of Radiological Sciences). Therefore, there are significant uncertainties in the RBE-value estimates. Furthermore, there are limited data on the RBE of the normal tissue. It is important to assess RBE values that are specific to tumors and normal tissues for various particle beams because normal tissue surrounds tumors and is occasionally included in the high-LET volume (e.g., a distal region) of carbon-ion irradiation.

Previous meta-analyses have established that the RBE value in tumor cell lines is not greater than that in normal-tissue-derived cell lines (normal cell lines). However, for the same LET, there are less obvious and distinguishable differences in radiosensitivity between tumor and normal cells [26]. To the best of our knowledge, there is a lack of available data on the difference in RBE values between several types of sarcoma cell lines on which particle therapy is effective (e.g., osteosarcoma, fibrosarcoma, chondrosarcoma, and liposarcoma) and normal cell lines, for which RBE values are generally difficult to derive because of their lower proliferation. To reduce uncertainty, different estimations of RBE should be performed using the same experimental protocols. Here, we present the results of a single-laboratory study that used a consistent experimental protocol to derive RBE for various sarcomas and normal cell lines in both proton and carbon-ion beams.

## 2. Results

### 2.1. Survival Fractions for Various Sarcoma Cell Lines for Gamma Ray, Proton Beam, and Carbon-Ion Irradiation

Although there is an existing open access database of α- and β-values in LQMs for various heavy ions as well as photon beams [25], which provides RBE values for various tumors, those data have large deviations, even in the same cell line, due to them being derived using different experimental protocols at different institutes. Our experiments were conducted using a consistent protocol at a single laboratory.

For the MG63 osteosarcoma cell line, a significantly greater cell-killing effect was observed with proton beam irradiation compared with that observed with gamma ray irradiation (Figure 1A). As shown in Figs. 1B–D, the survival fractions with both gamma ray and proton beam irradiation were almost identical in the HT1080 fibrosarcoma, SW872 liposarcoma, and SW1353 chondrosarcoma cell lines. With proton beam irradiation, the D_10_ doses (10% cell survival) for the MG63, HT1080, SW872, and SW1353 cell lines were 3.7, 6.8, 4.0, and 3.4 Gy, respectively (Table 1), and the corresponding RBE values were 1.29 ± 0.04, 0.97 ± 0.07, 1.04 ± 0.01, and 0.99 ± 0.02, respectively, for this endpoint (Table 1). The RBE value of MG63 was significantly greater than that of other sarcoma cell lines (Figure 2A).

Carbon-ion irradiation demonstrated the greatest cell-killing effect in all osteosarcoma cell lines (Figure 1). No shoulder was observed in their survival curves. For the D_10_ endpoint in MG63, HT1080, SW872, and SW1353 cells, the RBE value was estimated at 1.96 ± 0.23, 1.92 ± 0.16, 2.56 ± 0.32, and 1.94 ± 0.14, respectively (Table 1). In the SW872 cell line, the RBE value was markedly greater than that in other sarcoma cell lines. The other sarcoma cell lines had RBE values that were close to that of the HSG cell line used in a clinical treatment planning system at Hyogo Ion Beam Medical Center (HIBMC) (Figure 2B) [19].

### 2.2. Survival Fractions for Various Normal Cell Lines for Gamma Ray, Proton Beam, and Carbon-Ion Irradiations

Although RBE values are available for a number of tumor cell lines [25], there are limited data for normal cell lines. Survival fractions for normal cells irradiated with gamma ray, proton, and carbon-ion beams are shown in Figure 3. The hTERT-HME1 mammary gland cell line was radioresistant compared with other normal cell lines. The radiosensitivity of the HDF cell line was the highest with gamma ray and proton beam irradiations. However, with carbon-ion irradiation, the NuLi-1 cell line was more radiosensitive than HDF. Based on these survival curves, biological parameters were calculated for each cell line (α, β, D_10,_ and RBE) (Table 1). With proton irradiation, for the D_10_ endpoint, RBE-value estimates for the HDF, hTERT-HME1, and NuLi-1 cell lines were 1.09 ± 0.03, 1.05 ± 0.01, and 1.17 ± 0.06, respectively (Table 1). The RBE value of NuLi-1 was greater than that of hTERT-HME1 (Figure 4A); this difference was statistically significant. With carbon irradiation, RBE values for the HDF, hTERT-HME1, and NuLi-1 cell lines were 1.90 ± 0.03, 1.71 ± 0.03, and 2.54 ± 0.05, respectively (Table 1). The RBE value of NuLi-1 was significantly greater than that of the other cell lines (Figure 4B).

## 3. Discussion

We found a large heterogeneity among RBE values for different sarcoma and normal cell lines with different kinds of irradiation. The Particle Radiation Data Ensemble database version 3.2 developed by GSI provides substantially differing D_10_ and RBE values from different institutes, despite using the same cell lines. We conducted all our experiments using a consistent protocol at a single laboratory to directly compare the RBE-value dependency of tumor and normal tissue among gamma ray, proton beam, and carbon-ion irradiations.

Recent studies have reported that, in proton beam irradiation, LET is higher at the distal end of the SOBP than that at the center of SOBP, which leads to a higher RBE value at distal points [27,28,29]. Matsumoto et al. demonstrated that, in proton beam therapy, the RBE value of HSG cell line at the distal end of the SOBP was approximately 25% greater than that at the center of the SOBP [30]. They also demonstrated that the RBE value at the center of the SOBP was 1.20, which indicated that the radiosensitivity of proton beams at the SOBP center was higher than that of photon beams [30]. We did not investigate the cell type dependency of the RBE value at the distal point of the SOBP. However, our data revealed that the RBE value in MG63 osteosarcoma cells was 17% higher than 1.1, even at the SOBP center, which implies that a much higher RBE value could be expected at the distal point of the SOBP. These data suggest that proton therapy may biologically be more beneficial than photon beams for some types of tumors.

According to a report by Kagawa et al., the RBE values in the HSG cell line irradiated with carbon ions were 1.66, 1.76, and 2.27 at LETs of 39.6 keV/µm, 46.6 keV/µm, and 69.6 keV/µm, respectively [31]. The RBE value at an LET of 54.4 keV/µm can be estimated to be 1.95, based on a linear interpolation of the three values above. Our carbon-ion experiments performed at HIBMC using an LET of 54.4 keV/µm showed striking consistencies with the RBE values of the MG63, HT1080, and SW1353 cell lines and that of the HSG cell line (Figure 2B: red dotted line). In contrast, RBE values in the SW872 cell line were substantially greater than those of the HT1080 and HSG cell lines, suggesting that there are large variations between sarcoma cell lines. Chevalier et al. examined four different chondrosarcoma cell lines and demonstrated that they had heterogeneous radiosensitivity to carbon-ion beam irradiation [32], which suggests that a large heterogeneity in terms of radiosensitivity exists, even in the same kind of sarcoma. A possible reason for the variation in RBE values among cell lines may be differences in the expression of DNA repair pathway associated genes, such as Mer11 and Rad51. Chevalier et al. also reported that, compared with X-rays, carbon-ion irradiation induced a prolonged blockage of SW1353 cells in the G2 phase, a concomitant high level of γ-H2AX protein, and a higher incidence of micronuclei [32]. Further studies are necessary to identify the factors that characterize RBE values in carbon-ion irradiation. We have reported the successful use of artificial intelligence to identify carbon-ion-resistant cells in NR-S1 carcinomas [33]. This approach might also be used to characterize tumor and normal tissue dependence on RBE values in particle beam irradiation.

Reported RBE values for normal tissues have been determined using animal models [17]. In the present study, we performed colony assays to analyze the viability of normal tissues at the cellular level. Our result showed that hTERT-HME1 cells were radioresistant and had lower RBE values compared with other normal cell lines. The radiosensitivity of NuLi-1 cells was greater with both proton and carbon-ion beams, which indicates radiosensitivity dependence in normal cell lines. Moreover, a significantly higher RBE value was found between NuLi-1 and hTERT-HME1 cell lines in proton beam irradiation and between NuLi-1 and other two normal cell lines in carbon-ion irradiation, which suggests that the dependence of normal cells on RBE values should be carefully considered in clinical treatment planning systems.

With proton irradiation, the combined mean RBE value of the four sarcoma and three normal cell lines was close to 1.1; this indicates that the RBE of 1.1 utilized in current treatment planning systems is appropriate for calculating tumor and normal tissue doses [34]. However, in sarcoma cell lines, RBE values were heterogeneous, which should be considered in future treatment planning systems.

The RBE value in current carbon-ion treatment planning is based on the data of colony formation assay using HSG cells under aerobic conditions using the Heavy Ion Medical Accelerator in Chiba (HIMAC); this is because the RBE value of HSG cells was found to be close to that of other cancer cell lines [19,35]. Especially in current carbon-ion therapy, treatment planning systems have used an RBE value of 2.13 at an LET of 54.4 keV/µm. Our data showed that the average value of the RBE of four sarcoma and three normal cell lines was 2.08 ± 0.36, which is very close to the RBE value of 2.13. However, we observed wide-ranging RBE values with carbon-ion irradiation. In particular, RBE in SW872 and NuLi-1 increased by 34% and 48%, respectively, compared to that in HT1080 and hTERT-HME1, which clearly suggests that, for carbon irradiation, it is essential to evaluate specific doses for tumor and normal tissue. In this context, for some types of tumors (e.g., SW872 liposarcoma), dose reduction might be considered because of the high RBE value; an increased dose per fraction (i.e., hypofractionation) might be used to take advantage of the difference in RBE values between tumor and normal tissue (e.g., SW872 liposarcoma and HDF) [36]. However, caution should be exercised with regard to the dose for normal bronchus because of its high RBE.

Currently, a prescribed dose, determined on the basis of clinical experience, is used to compensate for uncertainties in the absolute RBE value [17]; this may improve tumor control and minimize normal tissue complications. Our results suggest that it is crucial to apply tissue-specific RBE values in current treatment planning systems, especially for normal tissue, where the RBE value differs greatly from that of tumor tissue. Our results support the approach employed by an ongoing clinical trial that uses different RBE calculation parameters and α- and β-values [37].

This study has some limitations. The RBE value, especially in tumor tissues, depends on several factors, including the speed of tumor growth and microenvironmental conditions, such as hypoxia. Although our cell survival experiments may have represented the intrinsic radiosensitivity of proliferating cells, the irradiated volume and the interactions in its substructure (e.g., the interactions of vascular and parenchymal cells) can immensely impact normal tissue. To resolve these issues, it will be necessary to conduct three-dimensional cell culture or in vivo experiments.

## 4. Materials and Methods

### 4.1. Cell Lines

We obtained HDF (human dermal fibroblast cell line) (Cat #, 106-05a) from CELL APPLICATIONS, INC. (San Diego, CA, USA). We purchased MG63 (human osteosarcoma), HT1080 (human fibrosarcoma), SW872 (human liposarcoma), SW1353 (human chondrosarcoma), hTERT-HME1 (human mammary epithelial cell line) (Cat #, CRL-4010), and NuLi-1 (human bronchial epithelial cell line) (Cat #, CRL-4011) from ATCC (Manassas, VA, USA).

HDF cells were maintained in basal medium with a growth supplement (Cat #, 115-485, 116-GS, CELL APPLICATIONS, INC., San Diego, CA, USA) at 37 °C in a humidified atmosphere containing 5% CO_2_. All sarcoma cell lines, hTERT-HME1, and NuLi-1 cells were cultured in Dulbecco’s Modified Eagle Medium (D-MEM) (Thermo Fisher Scientific, Waltham, MA, USA) containing 10% fetal bovine serum (Thermo Fisher Scientific) and 1% penicillin–streptomycin–glutamine mixed solution (nacalai tesque, Kyoto, Japan) under the same conditions (37 °C, 5% CO_2_). The hTERT-HME1 and NuLi-1 cells were immortalized by introducing the catalytic subunit of human telomerase (hTERT) or E6/E7 and hTERT, respectively.

### 4.2. Irradiation

Figure 5 shows the experimental setup for gamma ray (Figure 5A), proton beam (Figure 5B), and carbon-ion irradiations (Figure 5C) used in this study. The cells were irradiated with gamma rays, proton beams, and carbon ions. ^137^Cs gamma ray (0.662 MeV) irradiations were conducted using Gammacell^®^ 40 Exactor (MDS Nordion International Inc., Ottawa, ON, Canada), a gamma ray irradiator, at Osaka University Graduate School of Medicine. Proton beam irradiation was conducted at the Proton Beam Therapy Center of Hokkaido University. The cells were irradiated at the center of a 6 cm SOBP. For carbon-ion irradiation, the cells were irradiated at an LET of 54.4 keV/µm using a 250 MeV/nucleon carbon-ion beam at HIBMC. Dosimetry with a parallel plate ion chamber was conducted on the proton and carbon-ion beams before every irradiation. Gamma ray irradiation dosimetry was conducted using a Fricke dosimeter following the manufacturer’s instructions and confirmed using Gafchromic EBT3 films (Ashland Specialty Ingredients, Bridgewater, NJ, USA) by our group’s physicist.

### 4.3. Colony Formation Assay

Figure 5D shows the overall schedule of the colony formation assay. Cells were seeded onto 9.0 cm^2^ plastic slide flasks (Cat #, 170920, Thermo Fisher Scientific) 6 h before irradiation as described in a previous report [25]. Immediately after irradiation, the cells were washed in phosphate-buffered saline and trypsinized. The cells were then seeded onto 60 mm φ dishes. Approximately 2–3 weeks after culturing, the cells were fixed with 10% formalin and stained with crystal violet solution. Upon examination, colonies consisting of more than 50 cells were scored as survivors; the survival fraction (SF) was calculated using LQMs. All survival curves were fitted to LQMs using an in-house software with the scipy.optimize module run on Python (version 3.6.10) [38]. The LQM is expressed as “SF = exp (−αD−βD^2^)”, where D describes radiation dose (Gy). The parameters α and β are constants describing the liner component and quadratic part of the curve, respectively. A minimum of three independent experiments for each irradiation were performed.

### 4.4. Statistics

A two-tailed heteroscedastic t-test was performed to compare the RBE value of each cell line between proton beam and carbon-ion irradiations and to compare the SF between gamma ray and proton beams with a significance level of <0.05. Differences in α, β, and D_10_ of each cell line among gamma ray, proton beam, and carbon-ion irradiations and differences in RBE values among cell lines after radiation were analyzed using Tukey–Kramer honestly significant difference test with a significance level of <0.05, using the scikit-posthocs module run on Python (version 3.6.10) [39].

## 5. Conclusions

Our results showed a large RBE heterogeneity in various sarcoma and normal cell lines receiving carbon-ion and proton beams based on a consistent experimental protocol at a single laboratory. With proton irradiation, the mean RBE value of the four sarcoma and three normal cell lines was close to 1.1. This indicates that the RBE value of 1.1 utilized in current treatment planning systems is appropriate for calculating the doses for tumors and normal tissues. Proton therapy may be more beneficial for some types of tumors; however, caution must be exercised when considering normal tissues, which have higher RBE dependency. With carbon-ion irradiation, RBE values were heterogeneous in both sarcoma and normal cell lines. We observed wide-ranging RBE in carbon-ion irradiation, suggesting that it is important to evaluate tumor- and normal-tissue-specific doses. In this context, for some types of tumors, large heterogeneity in RBE values should bring into consideration dose reduction or an increased dose per fraction. However, one should be cautious with regard to the dose to normal tissues owing to their high RBE dependency. For tissue-specific dose evaluations in a treatment planning system, it is necessary to estimate the RBE values of relevant cell lines by acquiring their LET-dependent survival fractions.

## Figures and Tables

**Figure 1 cancers-14-02009-f001:**
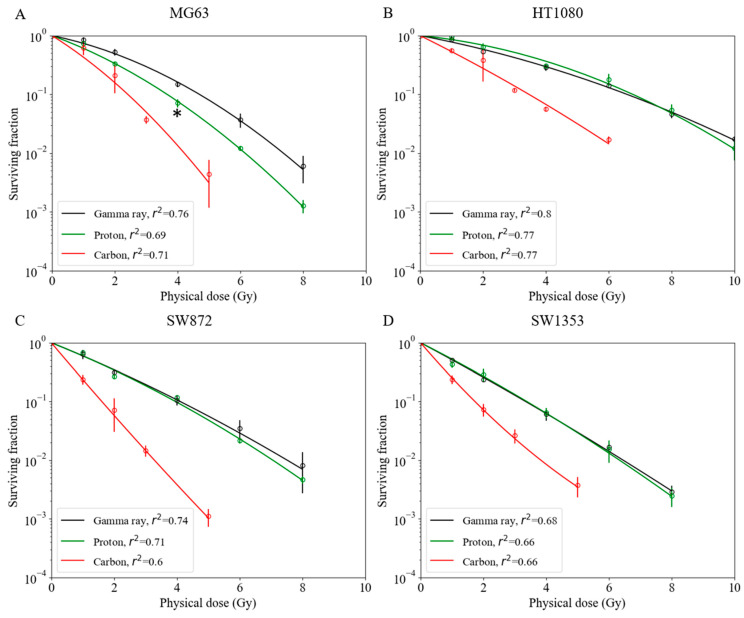
Survival fractions among various human sarcoma cell lines with gamma ray, proton beam, and carbon-ion irradiations. (**A**) MG63 osteosarcoma cell line; asterisk: * *p* < 0.05 (*t*-test). (**B**) HT1080 fibrosarcoma cell line. (**C**) SW872 liposarcoma cell line. (**D**) SW1353 chondrosarcoma cell line. Bars represent confidence intervals of one standard deviation. A minimum of three repeated experiments were performed for each irradiation method.

**Figure 2 cancers-14-02009-f002:**
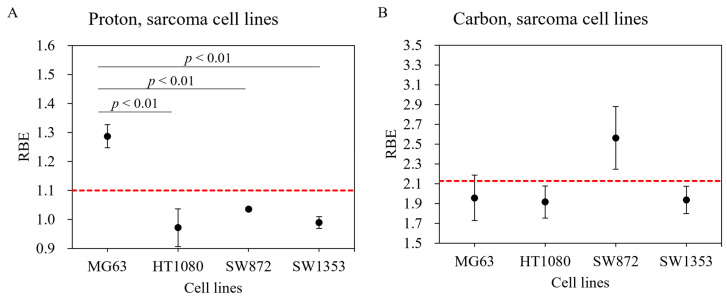
Relative biological effectiveness (RBE) of various human sarcoma cell lines exposed to proton beam and carbon-ion irradiations. Human sarcoma cell lines irradiated with (**A**) proton beam and (**B**) carbon ions. Bars represent confidence intervals of one standard deviation. The red dashed line indicates RBE values of 1.1 for proton beam and 2.13 for carbon-ion irradiations.

**Figure 3 cancers-14-02009-f003:**
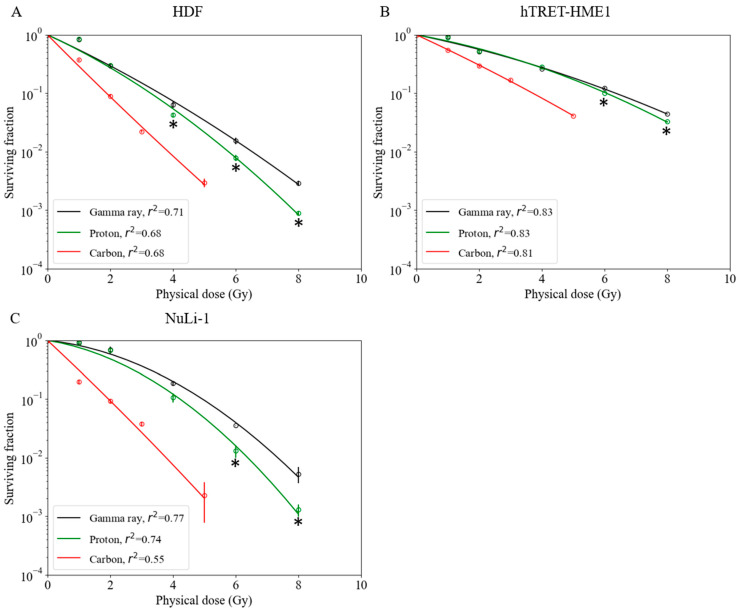
Survival fractions of normal cell lines irradiated with gamma rays, protons, and carbon ions. (**A**) Human dermal fibroblast cell line. (**B**) hTRET-HME1 human mammary epithelial cell line. (**C**) NuLi-1 human bronchial epithelial cell line. Bars indicate confidence intervals of one standard deviation. A minimum of three experiments were performed for each irradiation method. Asterisks indicate significant differences (* *p* < 0.05; *t*-test) between survival fractions after irradiation.

**Figure 4 cancers-14-02009-f004:**
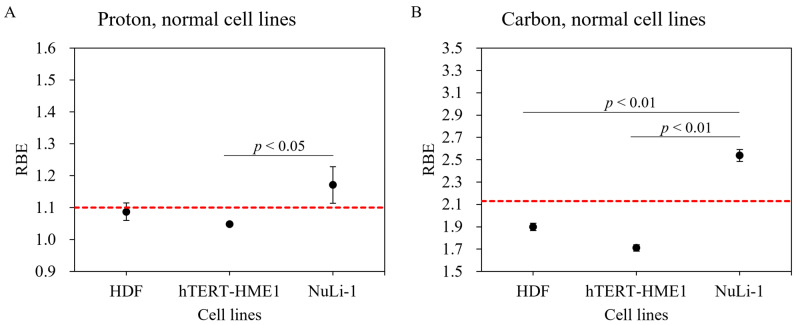
RBE values of various normal cell lines using proton and carbon-ion irradiation. Normal cell lines irradiated with (**A**) proton beam and (**B**) carbon ions. Bars represent confidence intervals of one standard deviation. The red dashed line indicates RBE values of 1.1 for proton beam and 2.13 for carbon-ion irradiations.

**Figure 5 cancers-14-02009-f005:**
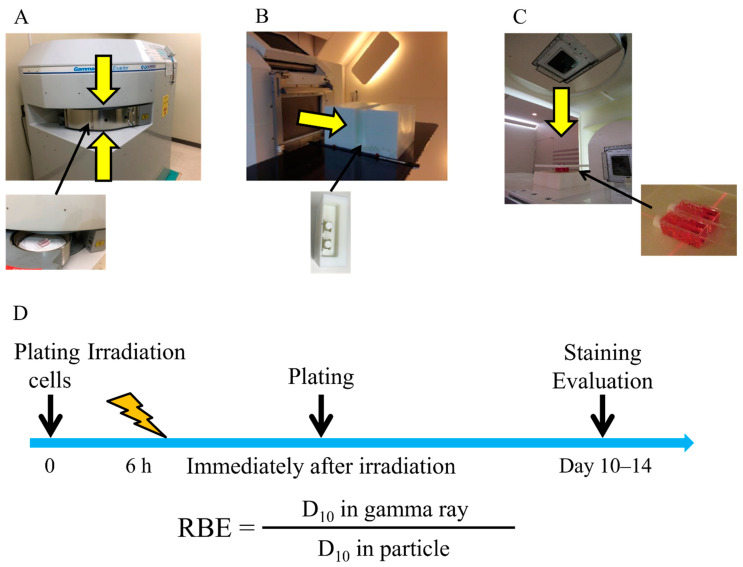
Experimental setup. (**A**) Experimental setup of gamma ray irradiation. (**B**) Experimental setup of proton beam irradiation at Hokkaido University. (**C**) Experimental setup of carbon-ion irradiation at Hyogo Ion Beam Medical Center. (**D**) Overall schedule of the colony formation assay.

**Table 1 cancers-14-02009-t001:** Biological parameters of various sarcoma and normal cell lines.

	Radiation	Sarcoma Cell Lines				Normal Cell Lines		
		MG63	HT1080	SW872	SW1353	HDF	hTERT-HME1	NuLi-1
α-value	Gamma ray	0.253 ± 0.0113	0.234 ± 0.0458	0.498 ± 0.115	0.660 ± 0.0448	0.568 ± 0.0506	0.254 ± 0.00973	0.133 ± 0.0750
	Proton	0.448 ± 0.0519	0.122 ± 0.0265	0.492 ± 0.0190	0.627 ± 0.0231	0.568 ± 0.0370	0.221 ± 0.0264	0.199 ± 0.0641
	Carbon	0.765 ± 0.149	0.607 ± 0.142	1.46 ± 0.272	1.43 ± 0.133	1.25 ± 0.0435	0.569 ± 0.0306	1.15 ± 0.0709
*p*-value	Gamma ray vs. Proton	<0.05	0.46	0.90	0.90	0.90	0.42	0.63
	Gamma ray vs. Carbon	<0.01	<0.05	0.12	<0.01	<0.01	<0.01	<0.01
	Proton vs. Carbon	<0.05	<0.01	0.12	<0.01	<0.01	<0.01	<0.01
β-value	Gamma ray	0.0503 ± 0.00598	0.0176 ± 0.00480	0.0153 ± 0.0242	0.00831 ± 0.00723	0.0211 ± 0.00749	0.0170 ± 0.00116	0.0672 ± 0.0130
	Proton	0.0488 ± 0.00852	0.0322 ± 0.00303	0.0228 ± 0.00376	0.0157 ± 0.00879	0.0396 ± 0.00499	0.0261 ± 0.00380	0.0817 ± 0.00457
	Carbon	0.0774 ± 0.0151	0.0165 ± 0.0219	−0.0165 ± 0.0587	−0.0606 ± 0.0323	−0.0134 ± 0.0143	0.0134 ± 0.00524	0.0184 ± 0.0369
*p*-value	Gamma ray vs. Proton	0.90	0.12	0.90	0.90	0.22	0.12	0.52
	Gamma ray vs. Carbon	0.12	0.77	0.77	<0.05	<0.05	0.64	0.28
	Proton vs. Carbon	0.28	0.77	0.77	<0.05	<0.01	<0.05	0.12
D_10_(Gy)	Gamma ray	4.72 ± 0.259	6.58 ± 0.230	4.10 ± 0.145	3.35 ± 0.128	3.58 ± 0.124	6.36 ± 0.0321	4.94 ± 0.0975
	Proton	3.67 ± 0.114	6.80 ± 0.437	3.96 ± 0.0258	3.39 ± 0.0696	3.29 ± 0.0824	6.06 ± 0.0371	4.23 ± 0.214
	Carbon	2.44 ± 0.298	3.46 ± 0.310	1.62 ± 0.204	1.74 ± 0.129	1.89 ± 0.0321	3.72 ± 0.0614	1.95 ± 0.0402
*p*-value	Gamma ray vs. Proton	<0.05	0.78	0.62	0.90	<0.05	<0.01	<0.01
	Gamma ray vs. Carbon	<0.01	<0.01	<0.01	<0.01	<0.01	<0.01	<0.01
	Proton vs. Carbon	<0.01	<0.01	<0.01	<0.01	<0.01	<0.01	<0.01
RBE	Proton	1.29 ± 0.0396	0.972 ± 0.0652	1.04 ± 0.00674	0.989 ± 0.0202	1.09 ± 0.0271	1.05 ± 0.00640	1.17 ± 0.0572
	Carbon	1.96 ± 0.229	1.92 ± 0.162	2.56 ± 0.318	1.94 ± 0.138	1.90 ± 0.0327	1.71 ± 0.0280	2.54 ± 0.0531
*p*-value	Proton vs. Carbon	0.05	<0.01	<0.05	<0.01	<0.01	<0.01	<0.01

## Data Availability

The data presented in this study are available on request from the corresponding author.

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
