# Peer review of "A Consistent Protocol Reveals a Large Heterogeneity in the Biological Effectiveness of Proton and Carbon-Ion Beams for Various Sarcoma and Normal-Tissue-Derived Cell Lines"

_cancers, 2022, doi:10.3390/cancers14082009_

Round 1

Reviewer 1 Report

Overall this revised manuscript is significantly improved. I believe that this work will be a great contribution to heavy charged particle radiobiology community.  Authors have addressed all of my concerns. I have no additional comments. The revised manuscript is ready for publication.

Author Response

To Reviewer 1

Overall this revised manuscript is significantly improved. I believe that this work will be a great contribution to heavy charged particle radiobiology community. Authors have addressed all of my concerns. I have no additional comments. The revised manuscript is ready for publication.

(Response)

Thank you for your review. We immensely appreciate your contribution to this review.

Reviewer 2 Report

The paper is presenting the RBE values of different cancer and normal cell lines using the single protocol for proton and carbon-ion beams. This data can be helpful for clinical application but needs major corrections:

  1. Please mention the novelty of your research. it has been little explanation in the introduction but must be elaborated
  2. The introduction is too short. please explain the others' works and explain them
  3. this sentence is not complete: Because no data are available of the difference of RBE between a large number of sarcoma and normal cell lines, especially by a consistent experimental protocol from a single laboratory. please rewrite it
  4. The English need editing
  5. in the results: Fig. 1A must come before Figure 1B-C
  6. Figure 1D is not mentioned in the text.
  7. the conclusion is too short.
  8. in this sentence: e.g. Lawrence Berkeley Laboratory, Gesellschaft für Schwerionenforschung mbH (GSI), National Institute of Radiological Sciences, etc. what does etc. mean? it's doesn't seem scientific

Author Response

Thank you for your review. We responded your comments and questions. Please see the attachment.

Round 2

Reviewer 2 Report

The paper has been completely improved and is ready for publication in Cancers.

This manuscript is a resubmission of an earlier submission. The following is a list of the peer review reports and author responses from that submission.

Round 1

Reviewer 1 Report

  1. Please delete the "Tissue" from the title.
  2. The English language needs to be edited in the whole paper.
  3. The abstract must be rewritten. Especially, the results part is a bit confusing. 
  4. Please include the novelty of your work in the introduction part.
  5. in "

    a consistent protocol in a single laboratory"

    What do authors exactly mean by protocol? Do you mean the protocol of colony assay?
  6. Figures are not referred to in the text. The authors have to explain every single figure in the results section.
  7. 2.1 Survival fractions for normal cell lines exposed to gamma ray, proton, and carbon-ion
  8.  should be rewritten.
  9. "Recent studies demonstrated that LET is higher at the distal.." in discussion: Have you investigated the RBE at both distal and center? If not, what is the point of mentioning this in the discussion section?
  10. "One of the possible reasons for this variation may be differences in biological char- 75 acteristics such as DNA repair pathway-associated gene expression changes caused by 76 carbon-ion irradiation..." in discussion: Please explain your own results and discuss the probable reasons.
  11. Please mention the references for" In the literature, RBE for the normal tissues was determined using animal models" in the discussion section.
  12.  The authors claimed "We developed a colony assay to analyze the normal tissue survival" but they have just used the common colony assay. please explain this.
  13. "In vivo study will necessary to solve these issues..": not necessarily. 3D cell culture would mimic the hypoxia situation. Besides that, for carbon and proton irradiation, hypoxia does not matter.
  14. 4.2 irradiation:  Specify which part of figure 5 is showing each treatment: 5A: gamma radiation,…
  15. The conclusion is the repetition of the introduction's last paragraph. Please rewrite it and include your suggestions and future perspective..

Reviewer 2 Report

In this manuscript, the authors used both cancer and normal tissue cell lines to systematically investigate the relative biological effectiveness (RBE) in carbon-ion and proton beams. The results reveal a large heterogeneity of RBE across these cancer and normal tissue cell lines in both carbon-ion and proton beams, which suggesting that tumor and normal tissue specific dose evaluation is necessary for proton and carbon-ion therapy. Overall, the study is very important, and the results are interesting. However, some major concerns are required to address before acceptance of publications.

  1. For the survival fraction study in all cell lines, although the authors showed averages and standard deviations in Table 1 and Figures 1-4, these averages and standard deviations were generated from replicates in one experiment or from repeated experiments? How many replicates are there at each dose for each experiment? How many times did the authors repeat their experiments in this study? If the experiments were not repeat, my suggestion is that the authors should repeat the experiments to show the reproducibility of their results. The authors should clarify all these information in their method.
  2. In Table 1, the authors should do statistical analysis of α, β, D10 and RBE between different treatment groups (gamma ray, proton and carbon ion beams) to assess whether they are statistically significant. The p-values should be included in the table.